

# Performance of interferon-γ release assay in the diagnosis of tuberculous lymphadenitis: a meta-analysis

Qianqian Liu[1,2,*], Wenzhang Li[3,*], Yunfeng Chen[1], Xinmiao Du[2], Chengdi Wang[2], Binmiao Liang[2], Yin Tang[4,5], Yulin Feng[2], Chuanmin Tao[6] and Jian-Qing He[2]

[1] Department of Respiratory Diseases, Chengdu Municipal First People's Hospital, Chengdu, Sichuan, China
[2] Department of Respiratory and Critical Care Medicine, West China Hospital, Sichuan University, Chengdu, Sichuan, China
[3] Department of Cardiology, First Affiliated Hospital of Chengdu Medical College, Chengdu, Sichuan, China
[4] State Key Laboratory of Oral Disease, West China School & Hospital of Stomotology, Sichuan University, Chengdu, Sichuan, China
[5] Herman Ostrow School of Dentistry, University of Southern California, Los Angeles, CA, USA
[6] Department of Laboratory Medicine, West China Hospital, Sichuan University, Chengdu, Sichuan, China
[*] These authors contributed equally to this work.

Corresponding author
Jian-Qing He, jianqhe@gmail.com

## ABSTRACT

**Background**. The diagnostic values of interferon-gamma release assays (IGRA) in tuberculosis (TB) vary a lot with different site of infections, with especially higher sensitivities in chronic forms of TB such as tuberculosis of the lymph node. We conducted a meta-analysis to comprehensively evaluate the overall accuracy of diagnostic IGRA for tuberculous lymphadenitis.

**Methods**. Pubmed, Web of Science, EMBASE, Wanfang and CNKI databases up to February 17, 2017 were searched to identify published studies. The study quality was evaluated using the QUADAS-2 checklist. The pooled estimates of diagnostic parameters were generated using a bivariate random-effects model and summary receiver operating characteristic (SROC) curves were used to summarize global performance.

**Results**. A total of ten qualified studies, performed in Korea or China, including 1,084 patients, were enrolled in this meta-analysis. The pooled estimates of diagnostic accuracy were as follows: sensitivity, 0.89 (95% CI [0.85–0.92]); specificity, 0.81 (95% CI [0.77–0.83]); positive likelihood ratio (PLR), 4.25 (95% CI [2.79–6.47]); negative likelihood ratio (NLR), 0.16 (95% CI [0.12–0.22]); and area under the curve (AUC) was 0.93. According to subgroup analyses, studies conducted using QuantiFERON-TB, in Korean population and focusing on cervical lymphadenitis exhibited relative higher specificity while lower sensitivity. No evidence of publication bias was identified.

**Conclusions**. IGRA exhibits high diagnostic accuracy in tuberculous lymphadenitis. The diagnostic value of IGRA differed by different IGRA methods, ethnicity and lymphadenitis location. Our conclusion may be more applicable to population from TB prevalent areas.

## INTRODUCTION

Despite the advances in effective chemotherapy regimens and slight decrease of incidence, tuberculosis (TB) still remains as a great challenge to public health. According to the most recent World Health Organization (WHO) report, TB has surpassed acquired immune deficiency syndrome (AIDS), becoming the leading cause of death from infectious disease, with an estimate of 9.6 million new cases and 1.5 million deaths every year (*WHO, 2015*). Moreover, the decline of extrapulmonary tuberculosis (EPTB) is regarded as suboptimal, and the proportion of EPTB among total TB cases has been increasing annually (*Barry et al., 2012*; *Peto et al., 2009*; *Sandgren, Hollo & Van der Werf, 2013*). Among all EPTB, tuberculous lymphadenitis, manifested as part of systemic processes or a unique lesion localized to lymph node, is the most common form in many areas, accounting for around 20–50% of total EPTB cases (*Gonzalez et al., 2003*; *Ilgazli et al., 2004*; *Wiwatworapan & Anantasetagoon, 2008*). The characteristic local impaired presentation, such as spontaneous fistula to the skin, was noted in only 4–11% of tuberculous lymphadenitis cases, and those classic TB systemaic signs, such as prolonged fever, night sweats and anorexia et al. were also not frequent, compared with pulmonary TB. Thus, the clinical physical findings of tuberculous lymphadenitis are usually indistinguishable from lymphadenitis caused by other etiologies, posing great diagnostic challenges to clinician, and giving it great potential for delayed treatment (*Fontanilla, Barnes & Von Reyn, 2011*; *Handa, Mundi & Mohan, 2012*; *Yoon et al., 2004*).

Up until now, several diagnostic choices are available for tuberculous lymphadenitis. Among them, the radiographic findings are frequently inconclusive and are merely clues for diagnosis (*Brodie & Schluger, 2005*). A tuberculin skin test (TST) cannot distinguish mycobacterium tuberculosis (MTB) infection from non-tuberculous mycobacteria (NTM) infection, as well as form the Bacillus Calmette-Guerin (BCG) vaccine, thus limiting its diagnostic application. Microbiologic and pathologic methods are usually the last procedures of choice; however, they can only be performed through invasive operation. Furthermore, the paucibacillary nature of the sample with non-uniform distributed bacilli often decreased the sensitivity of microbiologic diagnostic tests (*Brodie & Schluger, 2005*). Also, the highly variable performance of rapid PCR assay and non-specificity of histological examination usually leave the diagnosis of tuberculous lymphadenitis unsettled (*Hirunwiwatkul et al., 2002*; *Manitchotpisit et al., 1999*).

In recent years, an immune-based blood assay, interferon- γ (IFN- γ) release assay (IGRA), has been introduced for the diagnosis of TB based on the detection of IFN- γ secreted by T cells stimulated by two MTB–specific antigens, early secretory antigenic target (ESAT)-6 and culture filtrate protein (CFP)-10. The advantage of IGRA over TST depends on its lack of cross-reactivity with BCG vaccine strains and most NTM species (*Berthet et al., 1998*; *Pai et al., 2014*; *Pai et al., 2007*). It is reported that the diagnostic value of the IGRA may differed by clinical courses and site of infections, with especially higher sensitivities in chronic forms of TB, such as lymph node TB (sensitivity 89%) than in acute forms of TB (i.e., TB meningitis, sensitivity 74%) (*Cho et al., 2011*). The relative higher sensitivities among tuberculous lymphadenitis may be explained by more efficient immune response

allowing for the detection of IFN- γ production in those patients (*Pai, Kalantri & Menzies, 2006*). Over the past few years, a growing number of studies investigating the diagnostic accuracy of IGRA for tuberculous lymphadenitis have come out with inconsistent results, with sensitivity ranging from 0.81 to 0.95 and specificity ranging from 0.52 to 0.96 (*Kim, Ko & Kim, 2013*; *Kim, Kim & Woo, 2016*; *Liao et al., 2009*). The contradiction of previous studies may be partly due to small sample size of individual studies which are lack of statistical power. In order to get robust evidence guiding clinicians on the diagnostic accuracy of IGRA in the diagnosis of tuberculous lymphadenitis, we conducted the present meta-analysis.

## MATERIALS & METHODS

### Publication search

Five databases were searched from their inception until February 17, 2017: Pubmed, Web of Science, EMBASE, Wanfang and CNKI. The search strategy included the following terms: 'elispot' OR 'quantiferon\*' OR 'interferon-gamma release assay' OR 'interferon-gamma release assays' OR 'interferon-gamma releasing assay' OR 'interferon-gamma releasing assays' OR 't cell assay' OR 't cell assays' OR 'IGRA\*' OR 'T.SPOT\*' OR 'TSPOT\*' OR 'T-SPOT\*' in combination with 'lymphatic' OR 'lymph nodes' OR 'lymph node' OR 'lymphadenitis' in combination with 'diagnos\*' OR 'sensitivity'. Furthermore, a manual search for relevant studies in reference list of identified articles was also conducted. No language restrictions were applied in our study. Studies were included in this meta-analysis if they met the following criteria: (1) IGRA was used to test for the diagnosis of tuberculous lymphadenitis; (2) diagnostic $2 \times 2$ tables (the value of true positive, false positive, false negative, and true negative) were reported or could be calculated from original articles; (3) bacteriological findings or clinical response to anti-tuberculosis therapy were used as reference standard test. The following studies were excluded: (1) review or meta-analysis; (2) studies available only as abstracts; (3) duplicate publication of previous research.

### Data extraction

Two reviewers (Qianqian Liu and Wenzhang Li) independently screened those identified articles by titles and abstracts. Then the full texts were evaluated for eligibility based on pre-set inclusion and exclusion criteria. Discrepancies were resolved by discussion. The following variables were extracted from all included studies: first author, language, publication year, country, sample size, IGRA methods, IGRA cut-off values, and diagnostic $2 \times 2$ table values. Indeterminate results may occur in both IGRA tests and excluding these patients may lead our results to increased sensitivity and specificity. Therefore, those patients with indeterminate results were regarded as negative results in all analyses.

### Quality assessment

The revised Quality Assessment of Diagnostic Accuracy Studies (QUADAS-2) was used to assess the study quality focusing on the risk of bias and the applicability (*Whiting et al., 2011*). Any discrepancies by two reviewers were resolved by consensus.

**Table 1  Characteristics of studies included in meta-analysis.**

| First author | Language | Year | Country | IGRA method | Sample size | TP | FP | FN | TN |
|---|---|---|---|---|---|---|---|---|---|
| Kim KH | English | 2016 | Korea | QuantiFERON-TB | 244 | 34 | 9 | 8 | 193 |
| Kim YK | English | 2011 | Korea | QuantiFERON-TB | 108 | 25 | 13 | 2 | 68 |
| Song KH | English | 2009 | Korea | QuantiFERON-TB | 44 | 18 | 3 | 3 | 20 |
| Jia H | English | 2016 | China | T-SPOT.TB | 365 | 75 | 79 | 8 | 203 |
| Cho OH | English | 2011 | Korea | T-SPOT.TB | 64 | 48 | 4 | 6 | 6 |
| Liao CH | English | 2009 | China | T-SPOT.TB | 25 | 19 | 1 | 1 | 4 |
| Shin JA | English | 2015 | Korea | QuantiFERON-TB | 16 | 9 | 1 | 2 | 4 |
| Kim JK | Korean | 2013 | Korea | QuantiFERON-TB | 43 | 17 | 11 | 3 | 12 |
| Lu X | Chinese | 2016 | China | T-SPOT.TB | 19 | 14 | 1 | 2 | 2 |
| Jia HY | Chinese | 2014 | China | T-SPOT.TB | 156 | 47 | 22 | 4 | 83 |

**Notes.**

IGRA, interferon-$\gamma$ release assay; FN, false negative; FP, false positive; TN, true negative; TP, true positive.

## Statistical analysis

Heterogeneity was assessed by the Chi-squared test and $I^2$ test, setting $P < 0.10$ as the threshold for significance. The following accuracies with their 95% confidence interval (CI) were pooled respectively: sensitivity, specificity, positive likelihood ratio (PLR), negative likelihood ratio (NLR), using bivariate random-effects model. Summary receiver operating characteristic (SROC) curves were also generated. The Spearman rank correlation was conducted to test for potential threshold effect, setting $P < 0.05$ as threshold for significance. Presence of publication bias was assessed by Deeks' funnel plot with $P < 0.10$ for the slope coefficient suggested significant bias (*Deeks, Macaskill & Irwig, 2005*). All statistical tests were two-sided. Stata 12.0 (StataCorp, College Station, TX, USA) and Meta-DiSc 1.4 were used for statistical analysis.

# RESULTS

## Study characteristics

We initially identified 2,875 relevant articles and a total of ten studies meeting inclusion criteria were included in the final meta-analysis (Fig. 1) (*Cho et al., 2011; Jia et al., 2016; Jia et al., 2014; Kim, Ko & Kim, 2013; Kim, Kim & Woo, 2016; Kim et al., 2011; Liao et al., 2009; Lu et al., 2016; Shin et al., 2015; Song et al., 2009*). The basic characteristics of each study are presented in Table 1. The ten studies were conducted from 2009 to 2016, including 1,084 patients distributed in China (four studies) and Korea (six studies) respectively. Two studies were published in Chinese language, one in Korean language and the remaining were in English. The adopted IGRA methods were QuantiFERON.TB Gold In Tube (QFT-GIT; Cellestis, Carnegie, VIC, Australia) in five studies and T-SPOT.TB (Oxford Immunotech, Abingdon, UK) in the other five studies.

## Quality assessments of the eligible studies

The quality assessments of the ten eligible studies are listed in Table 2. The risk bias of selection domain was categorized as high in two studies, as they excluded patients with clinical diagnosed tuberculous lymphadenitis, which may exaggerate diagnostic accuracy

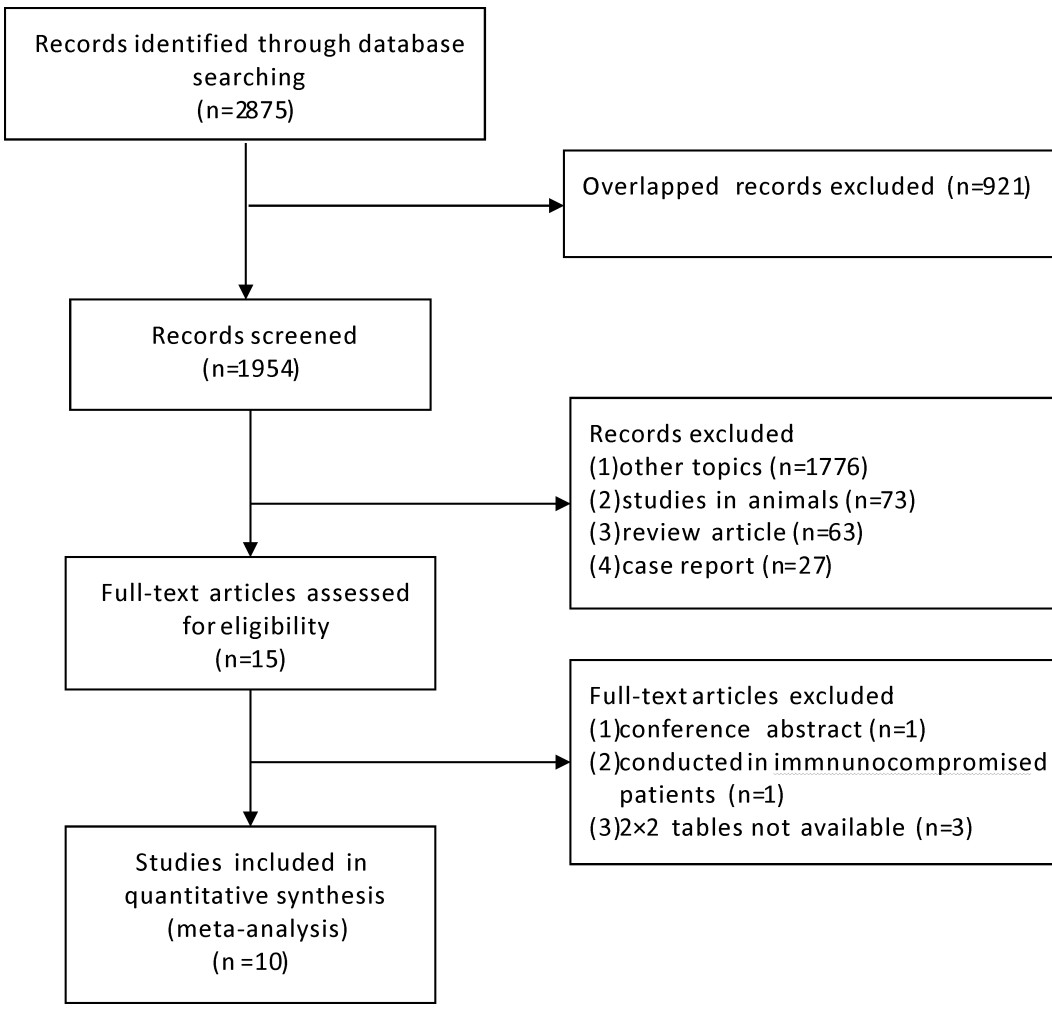

**Figure 1  Flow diagram of included studies.**

of index test. Only one study had low risk bias of index test as it was interpreted without knowledge of the other test, while the other studies did not report relevant information. As for the risk bias of references standard test, two studies had low risk, one study had high risk and the others were scored unknown. The risk bias of flow and timing domains and the concerns of applicability in all domains were categorized as low.

## Diagnostic performance

Chi-square values demonstrated no significant between-study heterogeneity among the following diagnostic parameters: sensitivity, 5.38 ($P = 0.80$) and NLR, 5.14 ($P = 0.82$), while significant heterogeneity was found for specificity ($P < 0.10$) and PLR ($P < 0.10$). The pooled estimates of diagnostic accuracy were listed as follows (Fig. 2): sensitivity, 0.89 (95% CI [0.85–0.92]); specificity, 0.81 (95% CI [0.77–0.83]); PLR, 4.25 (95% CI [2.79–6.47]); NLR, 0.16 (95% CI [0.12–0.22]). Figure 3 shows the SROC curve and the area under the curve (AUC) was 0.93.

**Table 2** Quality assessment of the eligible studies by QUADAS-2.

| Study | Risk of bias | | | | Applicability concerns | | |
|---|---|---|---|---|---|---|---|
| | Patient selection | Index test | Reference standard | Flow and timing | Patient selection | Index test | Reference standard |
| 1 | Unkown | Unknown | Unknown | Low | Low | Low | Low |
| 2 | Unkown | Unknown | Unknown | Low | Low | Low | Low |
| 3 | Unkown | Unknown | High | Low | Low | Low | Low |
| 4 | High | Low | Low | Low | Low | Low | Low |
| 5 | Unkown | Unknown | Low | Low | Low | Low | Low |
| 6 | Unknown | Unknown | Unknown | Low | Low | Low | Low |
| 7 | Unknown | Unknown | Unknown | Low | Low | Low | Low |
| 8 | Unknown | Unknown | Unknown | Low | Low | Low | Low |
| 9 | Unknown | Unknown | Unknown | Low | Low | Low | Low |
| 10 | High | Unknown | Unknown | Low | Low | Low | Low |

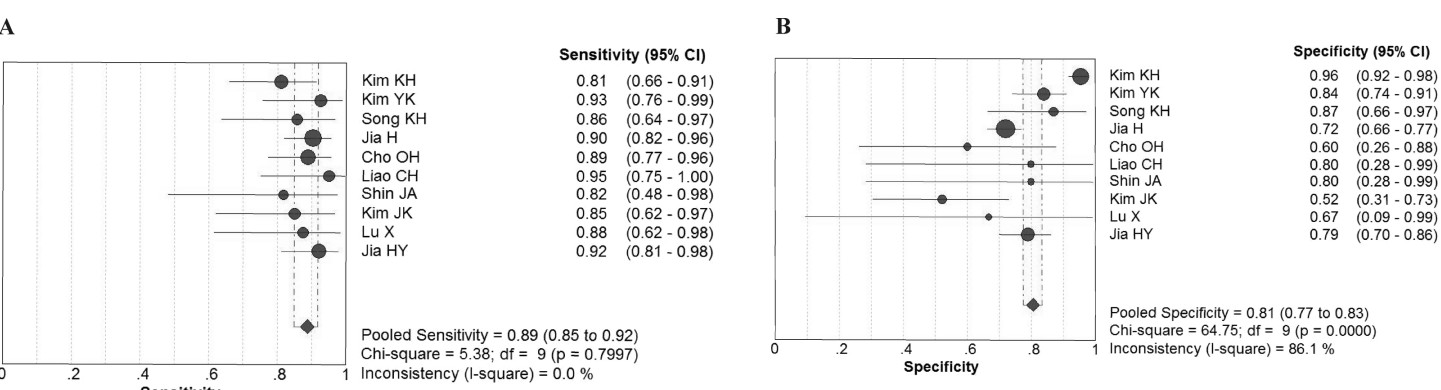

**Figure 2** Forest plot evaluating the sensitivity and specificity of diagnostic performance of IGRA in tuberculous lymphadenitis. CI, confidence interval. (A) Sensivitity; (B) Specificity.

## Threshold effect and subgroup analyses

If sensitivity and specicity were negatively correlated and ROC exhibited typical 'shoulder arm-shaped' distribution, then threshold effect should be considered. Our statistical results showed no significant threshold effect, with the spearman correlation coefficient to be 0.10 ($P = 0.78$).

Subgroup analyses by IGRA methods, ethnicity and tuberculous lymphadenitis locations were performed to identify potential sources of heterogeneity (Table 3). In general, the values of AUC in all subgroup analyses were more than 0.9 without exception. When stratified by IGRA methods, T-SPOT.TB had a relatively higher sensitivity and lower specificity than QuantiFERON-TB. While when stratified by ethnicity, studies conducted in Chinese population had a relatively higher sensitivity and lower specificity than those in Korean population. Some studies included patients with lymphadenitis localizing to cervical areas while other studies didn't restrict the location of lymph nodes, which may also lead to variability in diagnostic accuracy. Our subsequent subgroup analysis showed that limiting lymph nodes to cervical areas exhibited lower sensitivity and higher specificity compared with other studies.

**Table 3** Subgroup analyses.

| Subgroup | Studies | Sensitivity | | Specificity | | PLR | | NLR | | AUC |
|---|---|---|---|---|---|---|---|---|---|---|
| | (n) | Pooled value | $P_{\text{Heter}}$ | Pooled value | $P_{\text{Heter}}$ | Pooled value | $P_{\text{Heter}}$ | Pooled value | $P_{\text{Heter}}$ | Pooled value |
| T-SPOT.TB | 5 | 0.91(0.86–0.94) | 0.90 | 0.74(0.69–0.78) | 0.54 | 3.37(2.85–4.00) | 0.48 | 0.13(0.09–0.21) | 0.78 | 0.98(0.03) |
| QuantiFERON-TB | 5 | 0.85(0.78–0.91) | 0.73 | 0.89(0.85–0.92) | <0.10 | 5.53(2.11–14.52) | <0.10 | 0.19(0.12–0.29) | 0.72 | 0.92(0.02) |
| Korean | 6 | 0.86(0.80–0.91) | 0.77 | 0.88(0.84–0.91) | <0.10 | 4.71(2.08–10.67) | <0.10 | 0.19(0.13–0.28) | 0.84 | 0.92(0.02) |
| Chinese | 4 | 0.91(0.86–0.95) | 0.85 | 0.74(0.69–0.78) | 0.53 | 3.45(2.90–4.11) | 0.50 | 0.12(0.07–0.20) | 0.80 | 0.96(0.13) |
| Cervical TBL | 3 | 0.83(0.73–0.90) | 0.86 | 0.91(0.86–0.94) | <0.10 | 5.89(1.08–31.93) | <0.10 | 0.21(0.13–0.33) | 0.77 | 0.91(0.03) |
| Not limit to Cervical TBL | 7 | 0.90(0.86–0.94) | 0.92 | 0.75(0.71–0.79) | 0.29 | 3.72(2.97–4.66) | 0.29 | 0.14(0.09–0.20) | 0.82 | 0.95(0.04) |

**Notes.**

PLR, positive likelihood ratio; NLR, negative likelihood ratio; AUC, area under the curve; TBL, Tuberculous lymphadenitis.

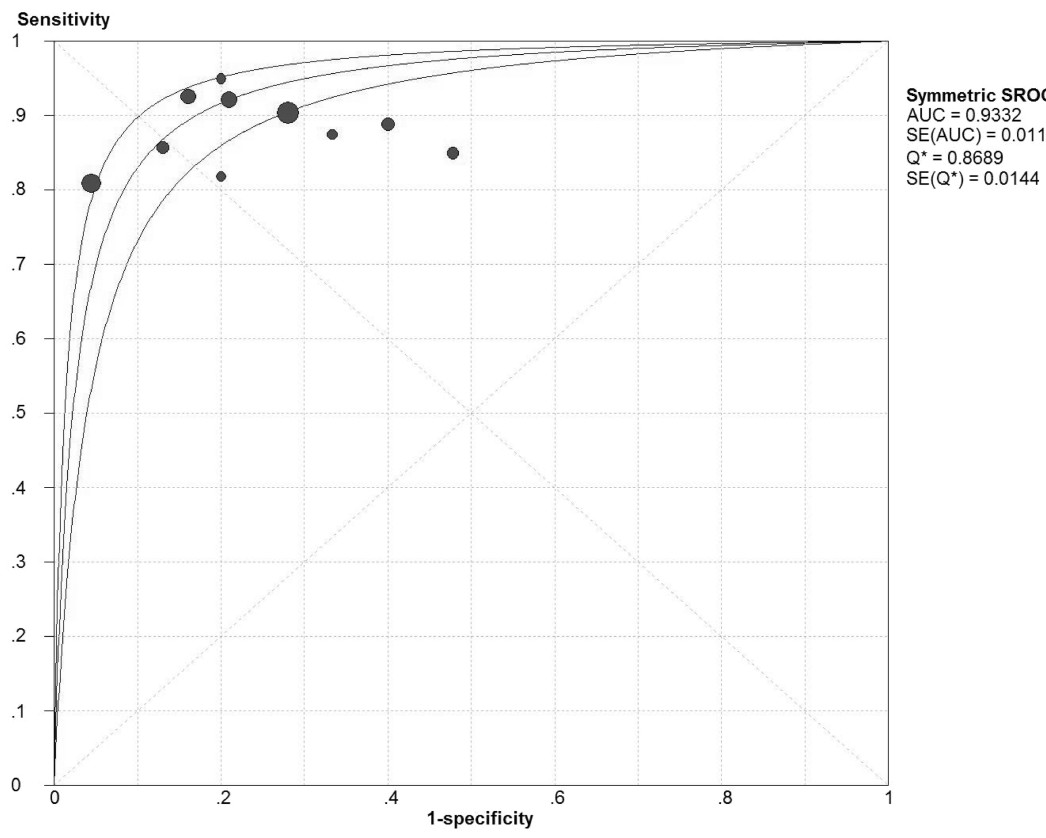

**Figure 3  Summary receiver operator characteristics (SROC) of IGRA on summary estimates of sensitivity and specificity.** AUC, area under the curve; SE, standard error

## Publication bias

Deeks' funnel plots was symmetric and the bias coefficient was not significant ($P = 0.54$) (Fig. 4). Our evaluation found no evidence of publication bias.

## DISCUSSION

The IGRA, an *in vitro* immunodiagnostic test, is being adopted increasingly for the detection of MTB infection and has generated promising results as an alternative diagnostic tool for the LTBI (*Diel et al., 2011*). However, the application of this newly emerging technique may not only be exclusive to LTBI. Several recent studies have investigated the diagnostic value of IGRA upon EPTB according to different sites of infection and uniformly reported a higher sensitivity of IGRA in tuberculous lymphadenitis detection than in other form of EPTB (*Cho et al., 2011*; *Song et al., 2009*). Thus, we conducted the present meta-analysis to comprehensively evaluate the overall diagnostic accuracy of IGRA upon tuberculous lymphadenitis.

The pooling results showed that the sensitivity and specificity of IGRA was 0.89 and 0.81 respectively, which means that 11% patients would have a missed diagnosis of tuberculous lymphadenitis and 19% would be falsely diagnosed as tuberculous lymphadenitis if we used IGRA as the only diagnostic method. The pooled PLR and NLR of IGRA was 4.25 and 0.16

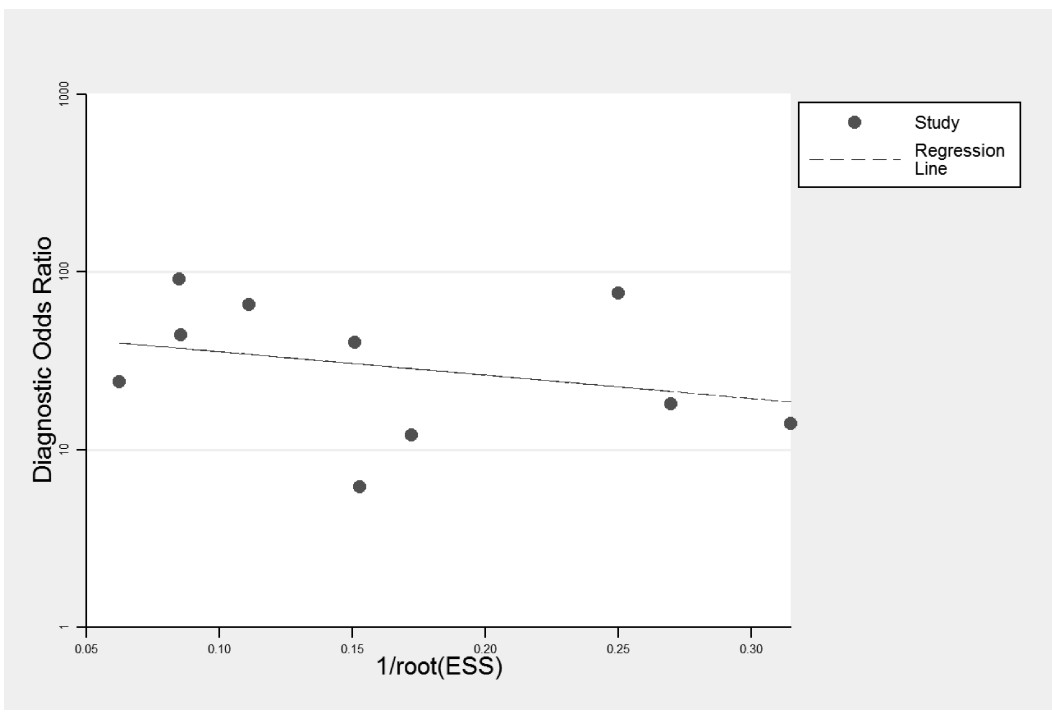

**Figure 4** Deeks' funnel plots for publication bias.

respectively, which indicates that patients with tuberculous lymphadenitis have 4.25 times the chance of getting a positive result than those without tuberculous lymphadenitis and those with tuberculous lymphadenitis have 0.16 times the chance of getting negative result than those without tuberculous lymphadenitis. AUC is a comprehensive index combining the sensitivity and specificity. Generally, AUC >0.9 is regarded to be highly accurate in diagnostic tests (*Deeks, 2001*). Accordingly, our summarized result of AUC (0.93) indicated high diagnostic accuracy. As meta-analysis is a pooling result of previous individual studies with larger statistical power, our conclusion is more reliable than previous small sample sized individual studies.

The subgroup results showed T-SPOT.TB had a relatively higher sensitivity and lower specificity than QuantiFERON-TB, which is in accordance with previous reports. Lee et al. conducted a head-to-head comparison between the two assays in the diagnosis of MTB infection. They reported higher sensitivity and inferior specificity of T-SPOT.TB over QuantiFERON-TB (*Lee et al., 2006*). The same tendency in diagnosing active tuberculosis was found by another study (*Lai et al., 2011*). It may be attributed to the fact that the QuantiFERON-TB assay tends to be more easily affected by immune status and peripheral lymphocyte counts or simply be related to the pre-set cut-off values of the two commercial assays (*Komiya et al., 2010*; *Lee et al., 2006*). Although we detected the same tendency with previous studies, our research focused on tuberculous lymphadenitis, which is different from previous studies not focusing on this special type of tuberculosis. When stratified by ethnicity, those studies conducted in the Chinese population had a relatively higher sensitivity and lower specificity than in Korean population. Such a similar tendency may

be partially explained by ethnic difference or by the fact that all studies conducted in the Chinese population of this meta-analysis were performed using T-SPOT.TB while all but one study conducted in the Korean population were performed using QuantiFERON-TB. Our subgroup analysis also showed that studies with lymphadenitis localizing to cervical areas exhibited lower sensitivity and higher specificity compared with studies which didn't restrict the location of lymph nodes. Whether there is a biological basis or it is just a statistical coincidence need to be further determined.

As for QUADAS-2 assessment, none of our included studies had low risk of bias in all domains. In patient selection domain, eight studies scored unclear because they didn't clearly state whether they enrolled sample of patients consecutively or randomly. What's more, in index test and reference standard test domains, most studies did not report whether the index test or reference standard test were interpreted without knowledge of the other test. It is important to note that low quality original research may introduce heterogeneity in meta-analysis. Thus, it needs to be stressed that the future diagnostic studies should be reported in detail according to relevant diagnostic study guidelines.

After all, some limitations should be pointed out. First, as most of patients included in the present study were immunocompetent, caution is advised when applying our conclusion to immunocompromised patients. Second, despite our thorough literature searching without language limitation, all studies of this meta-analysis were conducted either in Korea or in China; both were TB high prevalence areas. As we know, the diagnostic parameters, such as sensitivity and specificity, may be affected by the prevalence of the target disease. Thus, our conclusion might be more appropriate when used in TB high-burden areas.

## CONCLUSION

To sum up, IGRA exhibits high diagnostic accuracy in tuberculous lymphadenitis and may play an increasing important role as a supplementary diagnostic tool with the advantage of noninvasiveness and efficiency. The diagnostic value of IGRA differed by different IGRA methods, ethnicity and lymphadenitis location. Our conclusion may be more applicable to population from TB prevalent areas.

### Funding
The authors received no funding for this work.

### Competing Interests
The authors declare there are no competing interests.

### Author Contributions
- Qianqian Liu conceived and designed the experiments, performed the experiments, wrote the paper.
- Wenzhang Li conceived and designed the experiments, performed the experiments.
- Yunfeng Chen reviewed drafts of the paper.

- Xinmiao Du, Chengdi Wang and Jian-Qing He prepared figures and/or tables.
- Binmiao Liang and Yin Tang analyzed the data, contributed reagents/materials/analysis tools.
- Yulin Feng and Chuanmin Tao reviewed drafts of the paper.

## Data Availability

The raw data has been supplied as a Supplementary File.

## Supplemental Information

Supplemental information for this article can be found online at http://dx.doi.org/10.7717/peerj.3136#supplemental-information.

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
