# Peer review of "Performance of interferon-γ release assay in the diagnosis of tuberculous lymphadenitis: a meta-analysis"

_PeerJ, doi:10.7717/peerj.3136_

## Round 0.1 · original submission · Minor Revisions

· Academic Editor

Minor Revisions

Please revise according to the comments of the reviewers

Reviewer 1 ·

Basic reporting

No comments

Experimental design

No comments

Validity of the findings

No comments

Comments for the author

In this study, Liu et al. performed a meta-analysis to assess the overall accuracy of IGRA in the diagnosis of tuberculous lymphadenitis. The methodology was well performed and the results are interesting, however some issues need to be addressed before considering for publication.
1. The deadline of the literature search is June 30, 2016. Since it has been 5 months up to now, the authors should update their search.
2. In the introduction, line 57-58, the authors wrote that the laboratory findings of tuberculous lymphadenitis are usually indistinguishable from lymphadenitis. Please explain the reasons explicitly, otherwise the reader may be confused.
3. The abbreviation AUC should be explained at the first time when it appeared in the main text.
4. In the result section, line 150-154, the authors showed the P value of heterogeneity among the four diagnostic parameters twice, please delete any one of them and simplify the text.
5. The authors performed subgroup analyses by IGRA methods, countries and tuberculous lymphadenitis locations, I suggest the authors to conduct additional stratified analysis by quality sore since it may also contribute to the heterogeneity.
6. The English in the manuscript need to be improved and some grammatical errors should be corrected.

Reviewer 2 ·

Basic reporting

In the submission “Performance of interferon-γ release assay in the diagnosis of tuberculous lymphadenitis: a meta-analysis”, Qianqian Liu et al. evaluated the diagnostic accuracy of interferon-gamma release assays (IGRA) in the tuberculous lymphadenitis by a meta-analysis. They showed that IGRA exhibits high diagnostic accuracy in tuberculous lymphadenitis and their conclusion may be more applicable to population from TB prevalent areas. I suggest the authors revise their manuscript, and show their new findings (especially the results/advance beyond previous studies) explicitly in the revised manuscript.

Specific comments:
1. The authors stated in Introduction section that “Over the past few years, a growing number of studies investigating the diagnostic accuracy of IGRA for tuberculous lymphadenitis have arrived at inconsistent results, with sensitivity ranging from 0.81 to 0.95 and specificity ranging from 0.52 to 0.96. (Line 82, Page 10)”. However, they did not show how the contradiction in previous studies is induced. They should explicitly discuss about the contradiction in previous studies, and show what this study contributes to this issue.

2. A few literatures should be cited for sufficient detail & information to replicate. For example, in the Discussion section, “During subgroup analysis we also found that limiting lymph nodes to cervical areas exhibited lower sensitivity and higher specificity compared with other studies. (Line 204, Page 16)”

3. The authors should explicitly show their new findings, e.g., in the discussion or conclusion section, beyond previous studies. The authors stated in the discussion section that their results “T-SPOT.TB had a relatively higher sensitivity and lower specificity than QuantiFERON-TB, which is in accordance with previous reports. (Line 193, Page 16)”, “During subgroup analysis we also found that limiting lymph nodes to cervical areas exhibited lower sensitivity and higher specificity compared with other studies. (Line 204, Page 16)”. However, what new findings/ advance of current study are beyond previous results? This new findings/advance should explicitly show in abstract and conclusion section.

4. The table contents should be better sorted out (e.g. Table 1) and case-sensitive consistently (e.g. Table 2) to make it more legible.

Experimental design

no comment

Validity of the findings

no comment

---

## Round 0.2 · accepted · Accept

· Academic Editor

Accept

You have addressed all the concerns. Thanks for your submission.